Transperineal prostate biopsy guided by which ultrasound transducer: transrectal or transperineal: a retrospective study

Xiao Yang
Han Lina
Wang Han
Lyu Guorong
Li Shilin lslqz@fjmu.edu.cn
Department of Ultrasonography, Second Affiliated Hospital of Fujian Medical University , Quanzhou , Fujian , China
Zhang Xin
Electronic publication date: 2024 Nov 11
Publication date: 2024
Volume: 12
Electronic Location ID: e18424
Received 2024 Apr 15; Accepted 2024 Oct 8
Copyright: ©2024 Xiao et al.
Copyright year: 2024
Copyright holder: Xiao et al.
License: This is an open access article distributed under the terms of the Creative Commons Attribution License, which permits unrestricted use, distribution, reproduction and adaptation in any medium and for any purpose provided that it is properly attributed. For attribution, the original author(s), title, publication source (PeerJ) and either DOI or URL of the article must be cited.
License URL: https://creativecommons.org/licenses/by/4.0/

Keywords: Biopsy, Ultrasound guidance, Prostatic neoplasms, Detection rate, Complications

Funding: Joint funds for the innovation of science and technology, Fujian province No. 2023Y9231 This work was supported by the Joint funds for the innovation of science and technology, Fujian province (No. 2023Y9231). The funders had no role in study design, data collection and analysis, decision to publish, or preparation of the manuscript.

==============================
Background

Prostate biopsies are primarily conducted using either the transrectal or transperineal approach, with the ultrasound probe positioned in the rectum to obtain a clear view of the prostate. Reports on the utilization of transperineal prostate biopsies with the ultrasound probe placed on the perineal skin are limited.

Methods

A retrospective investigation was conducted on 119 patients who underwent transperineal ultrasound guided transperineal prostate biopsy (TP-TPPB). Additionally, 85 patients who underwent transrectal ultrasound guided transperineal prostate biopsy (TR-TPPB) were included as controls. The prostate cancer detection rates (PCDRs) and postoperative complication rates were compared between the two groups, and their application values were also evaluated.

Results

The overall PCDRs were 35.3% (42/119) in the TP-TPPB and 32.9% (28/85) in the TR-TPPB group (χ2 = 0.122, p = 0.727). When categorized by PSA level, there was no significant difference between the two groups in PCDRs in any category (p > 0.05). However, the single-needle PCDRs in some regions (L4, L5, R2, and apex) showed significant differences (p < 0.05). There was no difference in postoperative complication rates between the groups.

Conclusion

The PCDRs and the postoperative complication rates of TP-TPPB and TR-TPPB are similar. However, TP-TPPB requires simpler equipment and does not require intestinal preparation, which is especially useful for patients with rectal contraindications, such as those with severe hemorrhoids and rectal/anal diseases.

Introduction

Based on the 2020 report from the International Agency for Research on Cancer (IARC) (Sung et al., 2021), prostate cancer (PCa) affected 1.41 million men worldwide, making it the second most common cancer in males and the fifth leading cause of cancer-related deaths. Unlike many other tumors, PCa does not usually form a solid mass but rather develops diffusely throughout the prostate (Li et al., 2022). This dispersed nature makes the cancer more difficult to detect accurately in clinical settings, leading to many cases being diagnosed at an advanced stage or after metastasis has occurred (Miyahira et al., 2020; Welch & Albertsen, 2020). Additionally, PCa predominantly affects older men (Sung et al., 2021), many of whom have other chronic conditions. These comorbidities often complicate treatment options, resulting in poorer outcomes (Rycaj et al., 2017). Standard diagnostic tools for PCa include magnetic resonance imaging (MRI), prostate biopsy (PB) combined with prostate specific antigen (PSA) testing, and digital rectal examination (DRE) (Litwin & Tan, 2017). Among these, ultrasound guided PB is regarded as the most reliable method for diagnosing PCa.

The technique of freehand transperineal prostate biopsy (fTPPB) was first introduced in clinical practice in 1930. Although it was a significant advancement at the time, it had several drawbacks. The procedure was lengthy and had a low success rate due to the lack of imaging guidance, leading to patient discomfort and a higher risk of complications. In 1989, Kabalin et al. (1989) introduced the transrectal prostate biopsy (TRPB) technique, using ultrasound guidance for the first time. Over time, TRPB became the standard method due to its simplicity and efficiency (Omer & Lamb, 2019; Utsumi et al., 2021). However, as TRPB gained popularity, several limitations became apparent. The most concerning was the increased risk of infections and hematochezia, caused by the involvement of the rectum (Tops et al., 2022; Derin et al., 2020). While hematochezia usually resolves on its own, prostate infections can lead to bacteremia or sepsis, which can be life-threatening (Liss et al., 2011). Chronic prostatitis is another common issue post-TRPB, indicating long-term infection risks (Seo & Lee, 2018). Furthermore, the technique has lower sensitivity in detecting anterior PCa, which is a common site for cancer development (Stefanova et al., 2019). Given these challenges, transperineal prostate biopsy (TPPB) has recently regained attention as a viable alternative (Bhanji, Allaway & Gorin, 2021; Ong et al., 2015), particularly because of its higher detection rate for cancers in the anterior zone of the prostate  (Pepe et al., 2017).

Unlike its early use, modern TPPB is performed with real-time imaging guidance. Transrectal ultrasound guided transperineal prostate biopsy (TR-TPPB) has become widely used in recent years, offering a significant reduction in sepsis rates, in some cases bringing the risk close to zero (Pepe & Pennisi, 2022). However, the complex equipment required—such as biplanar intracavity transducers, guide frames, and templates—has limited its adoption in some healthcare settings. As a result, transperineal ultrasound guided transperineal prostate biopsy (TP-TPPB), also called perineal ultrasound guided transperineal prostate biopsy (PG-TPPB) in some studies (Xiao et al., 2024), has emerged as a simpler alternative. Despite its increasing use, there are still relatively few studies on the TP-TPPB techniques.

This study aims to explore the utility of TP-TPPB by evaluating its indications, procedure, clinical efficacy, and potential complications, thereby offering an alternative diagnostic tool for clinicians managing PCa.

Materials & Methods

A retrospective study was conducted involving 119 patients who had undergone TP-TPPB, along with an additional 85 patients who received TR-TPPB from August 1, 2020 to October 31, 2022. The Ethics Committee of the Second Affiliated Hospital of Fujian Medical University approved this study (approval code 233, date 2022), and all participants provided written informed consent.

The inclusion criteria encompassed the following: (1) detection of abnormal prostate nodules via DRE, regardless of PSA levels; (2) abnormal imaging findings, regardless of PSA levels; (3) PSA levels exceeding 10 ng/mL, regardless of free PSA and PSA density; and (4) PSA levels between 4 and 10 ng/mL with abnormalities in either free PSA or PSA density. Patients were excluded for any of the following reasons: (1) severe cardiovascular or cerebrovascular conditions; (2) impaired blood coagulation; (3) recent lower urinary tract surgeries; (4) urinary tract infections; (5) inability to comply with the procedure; and (6) incomplete medical or follow-up records. For TR-TPPB, patients with severe hemorrhoids or rectal/anal diseases were also excluded, leaving such patients to undergo TP-TPPB exclusively.

The color Doppler ultrasound equipment used during the procedures included the HI VISION Preirus (Hitachi, Tokyo, Japan) and the MyLab 8 eXP DH23 (Esaote, Genoa, Italy). TR-TPPB employed a TRT33 intracavitary transducer operating at 5–10 MHz, while TP-TPPB used a convex array B514 transducer, typically used for liver and kidney biopsies, operating at 2–5 MHz. Biopsies were performed using 18-gauge automatic biopsy guns (Bard Medical Technology, Shanghai, China), featuring a puncture slot of 20 mm and a sample diameter of 0.9 mm. Following an enhancement to the standard 10-core systematic biopsy protocol (Rudolph et al., 2020), an additional core was included to target the prostate apex, increasing the biopsy to an 11-core systematic biopsy. Standard 11-core systematic biopsies were performed (Fig. 1). Two experienced ultrasound physicians, each with over 5 years of experience, conducted the procedures, often working collaboratively. The biopsy samples were preserved in formalin and sent for pathological evaluation.

Before undergoing biopsies, all patients underwent auxiliary examinations, including MRI and color Doppler ultrasound to evaluate prostate size, volume, and location. Patients and their families were informed about the associated risks and gave written consent. The ultrasound physicians chose the appropriate biopsy technique based on individual patient conditions. Antibiotic prophylaxis, including oral metronidazole and gentamicin, was given 3 days prior to the biopsy for both patient groups. For TR-TPPB, a 300 mL normal saline enema was administered the day before the procedure, while no intestinal preparation was required for TP-TPPB. Patients remained still for 5–10 min during the procedures, and 10 min prior to the biopsy, a sterile field was created using 100 mL of 0.5% iodophor.

For TR-TPPB, patients were positioned in a lithotomy position with raised buttocks for better perineal exposure (Fig. 2A). The lower rectum, anus, perineum, and adjacent areas were disinfected, and 2% lidocaine was used for local anesthesia. The biopsies were performed under real-time rectal ultrasound guidance using 18-gauge automatic biopsy needles (Figs. 2B and 2C).

For TP-TPPB, patients were positioned similarly to TR-TPPB (Fig. 3A). The perineal region was sterilized using iodophor, and 2% lidocaine was administered for local anesthesia. Transperineal ultrasound was used to identify the optimal puncture sites, and the biopsy was performed under real-time perineal ultrasound guidance using 18-gauge automatic biopsy needles (Figs. 3B and 3C).

Figure 1 Puncture sites of standard 11 core systematic biopsies.

(A) Transverse plane. (B) Coronal plane. Pro, Prostate; Bl, Bladder; a, an apex of the prostate.

Figure 2 Schematic diagram of transrectal ultrasound guided transperineal prostate biopsy.

(A) TR-TPPB was performed with a lithotomy position and guided by an intracavitary TRT33 transducer. (B) Diagram of TR-TPPB in a sagittal plane. (C) Ultrasonogram of TR-TPPB in a sagittal plane. White arrow, needle tip; TR-TPPB, Transrectal ultrasound guided transperineal prostate biopsy; Bl, bladder; Re, rectal; Pro: prostate.

Figure 3 Schematic diagram of transperineal ultrasound guided transperineal prostate biopsy.

(A) TP-TPPB was performed with a lithotomy position and guided by a convex array B514 transducer. (B) Diagram of TP-TPPB in a sagittal plane. (C) Ultrasonogram of TP-TPPB in a coronal plane. White arrow, needle tip; TP-TPPB, transperineal ultrasound guided transperineal prostate biopsies; Bl, bladder; Re, rectal; Pro, prostate.

After the biopsies, the patients’ vital signs were monitored, and the final diagnoses were based on pathological results. Follow-up for all patients lasted at least six months. For those who underwent total or partial prostatectomy, the surgical pathology results were compared with the initial biopsy pathology. Any complications were documented. Infection was defined as a body temperature exceeding 38.5 °C with a positive blood culture, hematuria as visible blood in the urine within 3 days post-biopsy, hematochezia as rectal bleeding within 3 days after transrectal guidance procedure, and acute urinary retention as the sudden inability to urinate on the day of the procedure.

Statistical analysis

IBM SPSS Statistics for Windows, version 26.0 (IBM Corp., Armonk, NY), was used for data analysis. Continuous variables were evaluated for normality, with Student’s t-tests applied to compare normally distributed data, while nonparametric tests were used for data not following a normal distribution. Categorical data were expressed as case numbers (percentages), and the Chi-square test was used for comparisons between groups. A p-value of <0.05 was considered statistically significant.

Results

The basic and biopsy pathological characteristics of the patients are shown in Table 1. There were no significant differences in age (p = 0.846) and PSA (p = 0.806) between the TP-TPPB (n = 119) group and the TR-TPPB (n = 85) group. All biopsies were taken from the prostate and biopsy pathological diagnoses were acquired. In the TP-TPPB group, there were 42 cases of PCa, accounting for 35.3% of all cases, including 40 cases of acinar adenocarcinoma, one case of malignant epithelial tumor, and one case of neuroendocrine carcinoma. There were also eight cases of chronic prostatitis and 69 cases of benign prostatic hyperplasia. In the TR-TPPB group, there were 28 cases of PCa, accounting for 32.9% of all cases, including 27 cases of acinar adenocarcinoma, 1 case of intraductal carcinoma, 1 case of advanced intraepithelial neoplasia of the ductal epithelium, and 1 case of atypical small acinar hyperplasia (There were a small number of patients whose prostate biopsy pathological results indicated more than one pathological type). There were also 18 cases of chronic prostatitis and 39 cases of benign prostatic hyperplasia. Two of these patients had needle biopsies performed twice; their biopsy pathological results both indicated benign prostatic hyperplasia.

Table 1 Participant and biopsy pathological characteristics.

Demographics	TP-TPPB	TR-TPPB	P-value	
No.	119	85	–	
Mean ± SD age (yrs)	68.7 ± 10.7	68.4 ± 10.3	0.846	
ARa PSA (ng/mL)	103.36	101.30	0.806	
PCDRs (n, %)	42, 35.3%	28, 32.9%	0.727	
Acinar adenocarcinoma	40	27	–	
Malignant epithelial tumor	1	0	–	
Neuroendocrine carcinoma	1	0	–	
Intraductal carcinoma	0	1	–	
Advanced intraepithelial neoplasia of the ductal epithelium	0	1	–	
Atypical small acinar hyperplasia	0	1	–	
Chronic prostatitis	8	18	–	
Benign prostatic hyperplasia	69	39	–	
Notes.

a AR, Average Rank.

TP-TPPB, Transperineal ultrasound guided transperineal prostate biopsy.

TR-TPPB, Transrectal ultrasound guided transperineal prostate biopsy.

PSA, Prostate specific antigen.

PCDRs, Prostate cancer detection rates.

Some patients chose to undergo surgical total or partial prostatectomy, and intraoperative pathological results were as follows. In the TP-TPPB group, 20 cases received surgical treatment, resulting in 14 cases of benign prostatic hyperplasia and six cases of prostate acinar adenocarcinoma, which were completely consistent with the biopsy pathological results. In the TR-TPPB group, 34 cases received surgical treatment, resulting in 28 cases of benign prostatic hyperplasia and six cases of prostate acinar adenocarcinoma, which were almost completely consistent with the biopsy pathological results; one case had received a positive result following the needle biopsy (only R5 of all 11 needles) but this was incorrect based on an analysis of the intraoperative sample.

The overall PCDRs were 35.3% (42/119) in the TP-TPPB group and 32.9% (28/85) in the TR-TPPB group (χ2 = 0.122, p = 0.727). We then compared the PCDRs according to groupings based on PSA levels (Table 2). There was no significant difference between the two groups in the PCDRs of each level (p > 0.05 in all levels). The PCDRs in both groups, by PSA level, are shown (Fig. 4).

Table 2 PCDRs of different PSA values.

PSA (ng/mL)	TP-TPPB	TR-TPPB	χ 2	P-value	
≤4	33.3% (3/9)	16.7% (1/6)	–	0.604	
4–10	20.7% (6/29)	26.9% (7/26)	0.295	0.587	
10–20	19.0% (8/42)	26.1% (6/23)	0.119	0.730	
20–100	46.2% (12/26)	36.4% (8/22)	0.470	0.493	
>100	100.0% (13/13)	75.0% (6/8)	–	0.133	
Total	35.3% (42/119)	32.9% (28/85)	0.122	0.727	
Notes.

PCDRs, Prostate cancer detection rates.

PSA, Prostate specific antigen.

TP-TPPB, Transperineal ultrasound guided transperineal prostate biopsy.

TR-TPPB, Transrectal ultrasound guided transperineal prostate biopsy.

Figure 4 Comparison of PCDRs of different PSA level categories in the TP-TPPB group and TR-TPPB group.

PCDRs, Prostate cancer detection rates; PSA, Prostate specific antigen; TP-TPPB, Transperineal ultrasound guided transperineal prostate biopsy; TR-TPPB, Transrectal ultrasound guided transperineal prostate biopsy.

The single-needle positivity rates between the two groups are shown in Table 3 after grouping by the puncture site. The total single-needle PCDRs were 23.2% (304/1309) in the TP-TPPB group and 13.6% (127/935) in the TR-TPPB group (χ2 = 32.669, p < 0.001). However, subtle differences varied between the 11 puncture sites. There were significant differences in puncture sites of L4, L5, R2, and apex (L4: χ2 = 4.528, p = 0.033; L5: χ2 = 5.931, p = 0.015; R2: χ2 = 4.996, p = 0.025; apex: χ2 = 4.648, p = 0.031). There were no other significant differences (p > 0.05).

Table 3 PCDRs of single biopsy points.

PCDRs	TP-TPPB	TR-TPPB	χ 2	P-value	
L1	16.8% (20/119)	11.8% (10/85)	1.005	0.316	
L2	23.5% (28/119)	14.1% (12/85)	2.786	0.095	
L3	25.2% (30/119)	18.8% (16/85)	1.158	0.282	
L4	23.5% (28/119)	11.8% (10/85)	4.528	0.033a	
L5	28.6% (34/119)	14.1% (12/85)	5.931	0.015a	
R1	20.2% (24/119)	12.9% (11/85)	1.822	0.177	
R2	22.7% (27/119)	10.6% (9/85)	4.996	0.025a	
R3	23.5% (28/119)	12.9% (11/85)	3.595	0.058	
R4	21.8% (26/119)	15.3% (13/85)	1.378	0.240	
R5	24.4% (29/119)	14.1% (12/85)	3.245	0.072	
Apex	25.2% (30/119)	12.9% (11/85)	4.648	0.031a	
Total	23.2% (304/1309)	13.6% (127/935)	32.669	<0.001a	
Notes.

a The difference was statistically significant.

PCDRs, Prostate cancer detection rates.

TP-TPPB, Transperineal ultrasound guided transperineal prostate biopsy.

TR-TPPB, Transrectal ultrasound guided transperineal prostate biopsy.

Postoperative complications are shown in Table 4. All the complications were mild and responded to treatment. Fourteen cases, including 10 cases in the TP-TPPB group and four cases in the TR-TPPB group, had complications such as infections, hematuria, and urinary retention after the needle biopsy. The postoperative total complication rates did not significantly differ between TP-TPPB group and TR-TPPB group (8.4% vs 4.7%, χ2 = 1.061, p = 0.303). However, there were some differences in the types of complications. There was one case of infection in the TP-TPPB group but none in the TR-TPPB group (p > 0.05). There were 12 cases of hematuria, including eight in the TP-TPPB group and four in the TR-TPPB group (6.3% vs 4.7%, χ2 = 0.364, p = 0.546). There was one case of urinary retention in the TP-TPPB group but none in the TR-TPPB group (p > 0.05). There were no cases of hematochezia.

Table 4 Complications after PB.

Complications	TP-TPPB (n = 119)	TR-TPPB (n = 85)	χ 2	P-value	
Infections (%, n)	0.9%, 1	0%, 0	–	1.000	
Hematuresis (%, n)	6.3%, 8	4.7%, 4	0.364	0.546	
Hematochezia (%, n)	0%, 0	0%, 0	–	–	
Urinary retention (%, n)	0.9%, 1	0%, 0	–	1.000	
Total (%, n)	8.4%, 10	4.7%, 4	1.061	0.303	
Notes.

PB, Prostate biopsy.

TP-TPPB, Transperineal ultrasound guided transperineal prostate biopsy.

TR-TPPB, Transrectal ultrasound guided transperineal prostate biopsy.

Discussion

This study compared the newer TP-TPPB technique with the more established TR-TPPB technique, and intraoperative pathology results from both techniques closely matched the biopsy findings. This supports TP-TPPB as a viable alternative. The PCDRs and postoperative complication rates were found to be similar across both techniques. Furthermore, there was no significant difference in patient age or PSA levels between the groups, which helped to eliminate these variables as confounders in the study.

No significant differences in PCDRs were observed between the groups when analyzed by PSA levels. For both techniques, higher PSA levels were correlated with higher PCDRs, although an intriguing result emerged: the PCDRs in patients with PSA ≤ four ng/mL were higher than the latter two levels. This may be because biopsies were only performed after imaging or DRE identified prostate nodules. Additionally, single-needle PCDRs were calculated, showing TP-TPPB to be significantly higher than TR-TPPB, hinting that TP-TPPB may be more effective to some extent. This difference was particularly evident in regions like L4, L5, R2, and the apex. Notably, biopsies at apex points, which are hard to reach in traditional TRPB (Jiang et al., 2019), were more successful in TP-TPPB. In TR-TPPB, the PCDRs of apex points were also significantly lower than those of TP-TPPB. Does this mean that apex points are difficult to obtain when guided or punctured in the rectum? This is just a preliminary conjecture, and further investigation is needed to confirm these findings. The L4, L5, and R2 points may resemble the apex points or reflect sampling error.

Both techniques use perineal puncture, but the main distinction lies in the guidance method. TR-TPPB is guided through the rectum using a high-frequency biplanar intracavitary probe, providing clear images of the prostate. In contrast, TP-TPPB employs a low-frequency convex array probe for perineal guidance, often used in liver and kidney biopsies. While the image quality with perineal guidance is slightly lower than with rectal guidance, with the rectal guided probe closer to the prostate, and it being a high-frequency probe, which can display more clearly, it is still sufficient for the operator to perform systematic regional biopsies. Since PCa is generally distributed throughout the prostate (Li et al., 2022), the similar PCDRs between both techniques affirm their effectiveness.

Furthermore, TR-TPPB has some limitations. The templates required are costly, limiting its availability in some healthcare settings. Although templates enhance accuracy, they restrict flexibility in adjusting puncture angles, making it harder to achieve ideal puncture positions. Some clinicians bypass templates entirely (Szabo, 2021), but this demands greater skill and increases the risk of puncture failure and tissue damage due to the separation between the puncture site and probe. However, TP-TPPB does not require a template or intracavitary probe, needing only an abdominal probe. It can also be performed with or without a guide frame for puncture assistance, making it more adaptable in resource-limited areas. Additionally, as TP-TPPB avoids the rectum, there’s no need for intestinal preparation, making it a better option for patients with conditions like severe hemorrhoids or rectal/anal diseases.

There are some limitations to this study. The sample size and number of recorded complications were relatively small. Although the increase in sample size makes up for some shortcomings in the previous study (Xiao et al., 2024), a further investigation with larger cohorts is needed to verify its stability. Also, all patients received prophylactic antibiotics, whereas earlier studies (He et al., 2022) have shown no difference in infection rates between those receiving perineal punctures with and without antibiotics. Future research might consider excluding antibiotic use. Moreover, patients with severe hemorrhoids or rectal/anal conditions were only treated with TP-TPPB in this study, introducing potential selection bias. Further studies should involve patients without such conditions to validate these findings.

Conclusions

In conclusion, the PCDRs and the postoperative complication rates of TP-TPPB and TR-TPPB are similar. However, TP-TPPB requires simpler equipment and does not require intestinal preparation, which is especially useful for patients with rectal contraindications, such as patients with severe hemorrhoids and rectal/anal diseases.

Supplemental Information

Supplemental Information 1 Statistical data and statistical methods

Supplemental Information 2 STROBE checklist

The authors would like to express their gratitude to EditSprings for the expert linguistic services provided.

Additional Information and Declarations

Competing Interests

Author Contributions

Human Ethics

Data Availability

The authors declare there are no competing interests.

Yang Xiao conceived and designed the experiments, performed the experiments, analyzed the data, prepared figures and/or tables, authored or reviewed drafts of the article, and approved the final draft.

Lina Han performed the experiments, analyzed the data, authored or reviewed drafts of the article, and approved the final draft.

Han Wang performed the experiments, analyzed the data, authored or reviewed drafts of the article, and approved the final draft.

Guorong Lyu conceived and designed the experiments, analyzed the data, authored or reviewed drafts of the article, and approved the final draft.

Shilin Li conceived and designed the experiments, analyzed the data, prepared figures and/or tables, authored or reviewed drafts of the article, and approved the final draft.

The following information was supplied relating to ethical approvals (i.e., approving body and any reference numbers):

Ethics Committee of the Second Affiliated Hospital of Fujian Medical University (protocol code 233 and date of approval 2022).

The following information was supplied regarding data availability:

The raw data are available in the Supplementary File.

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
