# Peer review of "Transperineal prostate biopsy guided by which ultrasound transducer: transrectal or transperineal: a retrospective study"

_PeerJ, doi:10.7717/peerj.18424_

## Round 0.1 · original submission · Major Revisions

The authors are requested to carefully revise the manuscript and answer the questions raised by the reviewers.

Reviewer 1 ·

Basic reporting

No comment

Experimental design

No comment

Validity of the findings

No omment

Additional comments

The Discussion should be improved:

1. Data about the better accuracy of transperineal prostate biopsy in diagnosing prostate cancer should be added; in detail, the better detection rate for PCa located in the anterior zone should be reported (Pepe P, Garufi A, Priolo G, Pennisi M. Transperineal Versus Transrectal MRI/TRUS Fusion Targeted Biopsy: Detection Rate of Clinically Significant Prostate Cancer. Clin Genitourin Cancer. 2017 Feb;15(1):e33-e36. doi: 10.1016/j.clgc.2016.07.007). In addition, the advantages of transperineal biopsy in comparison with transrectal approach to reset the risk of sepsis should be added (Pepe P, Pennisi M. Morbidity following transperineal prostate biopsy: Our experience in 8.500 men. Arch Ital Urol Androl. 2022 Jun 29;94(2):155-159. doi: 10.4081/aiua.2022.2.155)
2. The Authors should explain why nobody permorfed MRI/TRUS fusion biopsy; multiparametric MRI was performed before prostate biopsy following international guidelines? In addition, is transperineal ultrasound-guided biopsy accurate to perform targeted biopsy?
3. Data regarding the emert role of PSMA PET/CT to perform targeted biopsy for the diagnosis of clinically significant PCa should be added (Pepe P, Pepe L, Cosentino S, Ippolito M, Pennisi M, Fraggetta F. Detection Rate of 68Ga-PSMA PET/CT vs. mpMRI Targeted Biopsy for Clinically Significant Prostate Cancer. Anticancer Res. 2022 Jun;42(6):3011-3015. doi: 10.21873/anticanres)
4. The unique advantage of transperineal ultrasound-guided biopsy (men submitted to abdomen-perineal surgery) should be underlined; on the other hand, the lower accuracy to perform targeted biopsy should be reported.
5. European Urological Guidelines recommend to perform transperineal prostate biopsy to reset the risk of sepsis

Reviewer 2 ·

Basic reporting

I read with interest the study entitled "Transperineal prostate biopsy guided by which ultrasound transducer: transrectal or transperineal?

I have several concerns:

- Regarding the methodology several concerns should be addressed as to why choosing only prostate cancer detection rate and not only clinically significant PCa as it is the outcome of interest in the urological community.

- The type of biopsy is obsolete as several guidelines recommend target +/- systematic

- The population has very high levels of PSA so it's not representative of the normal population who undergo prostate biopsies

- The results section is confused and results presented poorly (how is it possible that patients who underwent prostatectomy have benign pathology?)

Overall the study has a limited population and several methodological errors, also the topic is not very interesting

Experimental design

as above

Validity of the findings

as above

---

## Round 0.2 · Minor Revisions

The authors are requested to carefully revise the manuscript and answer the questions raised by the reviewers.

Reviewer 1 ·

Basic reporting

The manuscript has been improved

Experimental design

The study has been improved

Validity of the findings

Conclusion should be supported by a greater number of procedures

Reviewer 2 ·

Basic reporting

None of my comments were taken into consideration, neither the authors responded to them

Experimental design

-

Validity of the findings

-

Additional comments

-

Reviewer 3 ·

Basic reporting

The updated version of the manuscript is clearer with the inclusion of figures illustrating the TP-TPPB and TR-TPPB processes. However, I would still suggest that the authors include a dedicated section discussing the features and limitations of the current state-of-the-art technologies, particularly emphasizing the comparison between PG-TPPB, TP-TPPB, and TR-TPPB. This would help to contextualize the findings of both articles and clarify how these methods relate to each other.

Regarding the re-used text, I understand that the articles were part of a series of studies conducted simultaneously, and as such, some repetition is unavoidable. However, to minimize redundancy and improve the distinctiveness of each article, I recommend revisiting the introduction, methods, and discussion sections. Specifically, the authors could focus on highlighting the unique aspects of the TP-TPPB and TR-TPPB comparison in the current manuscript, while also addressing the specific advantages or limitations of PG-TPPB as discussed in the previously published article.

Experimental design

N/A

Validity of the findings

N/A

Reviewer 4 ·

Basic reporting

The reporting is fine with some minor grammar errors. It would be good that the author can proofread it before final submission.

Experimental design

It would be informative to include the enrollment periods for both the cases and controls. In addition, the authors should provide some clarifications on whether the number of controls are much less than the number of cases.

Validity of the findings

The authors mentioned that the biopsies were conducted by two physicians and failed to discuss the situation when the biopsy from each physician did not agree.
In Figure 4, is there a reason the PSA values are grouped into these categories?
In Table 3, the total row is hard to interpret, does it mean that it is a sum of all the single biopsy points? Does each patient has a single biopsy point results for L1-L5, R1-R5, and apex?

---

## Round 0.3 · Minor Revisions

I agree with the reviewer's opinion. For example, the schematic diagrams in Figures 1, 2, and 3 could have used more realistic anatomical illustrations instead of being hastily drawn like a child's sketches. I believe that as an influential international journal, such quality of images is unacceptable.

Reviewer 1 ·

Basic reporting

The manuscript has been improved

Experimental design

The manuscript has been improved

Validity of the findings

The manuscript has been improved

Additional comments

The manuscript has been improved

Reviewer 3 ·

Basic reporting

The revisions have significantly enhanced the manuscript. However, I would suggest improving the quality of the figures, as some of them appear too sketchy.

Experimental design

N/A

Validity of the findings

N/A

Reviewer 4 ·

Basic reporting

The authors have addressed comments from previous review. No additional comment.

Experimental design

The authors have addressed comments from previous review. No additional request.

Validity of the findings

The authors have addressed comments from previous review. No additional comment.

---

## Round 0.4 · accepted · Accept

After revisions, three reviewers agreed to publish the manuscript. I also reviewed the manuscript and found no obvious risks to publication. Therefore, I also approve the publication of this manuscript.